# Clinical Workflow of Cone Beam Computer Tomography-Based Daily Online Adaptive Radiotherapy with Offline Magnetic Resonance Guidance: The Modular Adaptive Radiotherapy System (MARS)

**DOI:** 10.3390/cancers16061210

**Published:** 2024-03-19

**Authors:** Ji-Young Kim, Bouchra Tawk, Maximilian Knoll, Philipp Hoegen-Saßmannshausen, Jakob Liermann, Peter E. Huber, Mona Lifferth, Clemens Lang, Peter Häring, Regula Gnirs, Oliver Jäkel, Heinz-Peter Schlemmer, Jürgen Debus, Juliane Hörner-Rieber, Fabian Weykamp

**Affiliations:** 1Department of Radiation Oncology, Heidelberg University Hospital, 69120 Heidelberg, Germany; ji-young.kim@med.uni-heidelberg.de (J.-Y.K.); b.tawk@dkfz.de (B.T.); m.knoll@dkfz-heidelberg.de (M.K.); philipp.hoegen@med.uni-heidelberg.de (P.H.-S.); jakob.liermann@med.uni-heidelberg.de (J.L.); p.huber@dkfz-heidelberg.de (P.E.H.); juergen.debus@med.uni-heidelberg.de (J.D.); juliane.hoerner-rieber@med.uni-heidelberg.de (J.H.-R.); 2Heidelberg Institute of Radiation Oncology (HIRO), 69120 Heidelberg, Germany; o.jaekel@dkfz-heidelberg.de; 3Clinical Cooperation Unit Radiation Oncology, German Cancer Research Center (DKFZ), 69120 Heidelberg, Germany; 4National Center for Tumor Diseases (NCT), 69120 Heidelberg, Germany; 5Clinical Cooperation Unit Translational Radiation Oncology, National Center for Tumor Diseases (NCT), Heidelberg University Hospital (UKHD) and German Cancer Research Center (DKFZ), 69120 Heidelberg, Germany; 6German Cancer Consortium (DKTK), Core Center Heidelberg, 69120 Heidelberg, Germany; 7Clinical Cooperation Unit Molecular Radiooncology, German Cancer Research Center (DKFZ), 69120 Heidelberg, Germany; 8Division of Medical Physics in Radiation Oncology, German Cancer Research Center (DKFZ), 69120 Heidelberg, Germany; m.lifferth@dkfz-heidelberg.de (M.L.); clemens.lang@dkfz-heidelberg.de (C.L.); p.haering@dkfz-heidelberg.de (P.H.); 9Division of Radiology, German Cancer Research Center (DKFZ), 69120 Heidelberg, Germany; r.gnirs@dkfz-heidelberg.de (R.G.); h.schlemmer@dkfz-heidelberg.de (H.-P.S.); 10Department of Radiation Oncology, Heidelberg Ion-Beam Therapy Center (HIT), Heidelberg University Hospital, 69120 Heidelberg, Germany

**Keywords:** online adaptive radiotherapy, MR-guided radiotherapy, Ethos

## Abstract

**Simple Summary:**

Daily adaptation of the radiation plan on the treatment couch has the potential to reduce toxicity to healthy tissue while maintaining the dose or even enabling higher doses applied to cancerous tissue. However, current approaches either require more time and personnel to perform plan adaptation or lack resolution for soft tissue contrast. The Ethos radiotherapy device utilizes artificial intelligence and machine learning to allow rapid plan adaptation based on the daily anatomy. We here outline the first workflow in which we combine the Ethos machine with weekly magnetic resonance imaging, the gold standard for soft tissue resolution. This is facilitated by a shuttle system that allows patients to be transported from one device to the other in treatment position.

**Abstract:**

Purpose: The Ethos (Varian Medical Systems) radiotherapy device combines semi-automated anatomy detection and plan generation for cone beam computer tomography (CBCT)-based daily online adaptive radiotherapy (oART). However, CBCT offers less soft tissue contrast than magnetic resonance imaging (MRI). This work aims to present the clinical workflow of CBCT-based oART with shuttle-based offline MR guidance. Methods: From February to November 2023, 31 patients underwent radiotherapy on the Ethos (Varian, Palo Alto, CA, USA) system with machine learning (ML)-supported daily oART. Moreover, patients received weekly MRI in treatment position, which was utilized for daily plan adaptation, via a shuttle-based system. Initial and adapted treatment plans were generated using the Ethos treatment planning system. Patient clinical data, fractional session times (MRI + shuttle transport + positioning, adaptation, QA, RT delivery) and plan selection were assessed for all fractions in all patients. Results: In total, 737 oART fractions were applied and 118 MRIs for offline MR guidance were acquired. Primary sites of tumors were prostate (*n* = 16), lung (*n* = 7), cervix (*n* = 5), bladder (*n* = 1) and endometrium (*n* = 2). The treatment was completed in all patients. The median MRI acquisition time including shuttle transport and positioning to initiation of the Ethos adaptive session was 53.6 min (IQR 46.5–63.4). The median total treatment time without MRI was 30.7 min (IQR 24.7–39.2). Separately, median adaptation, plan QA and RT times were 24.3 min (IQR 18.6–32.2), 0.4 min (IQR 0.3–1,0) and 5.3 min (IQR 4.5–6.7), respectively. The adapted plan was chosen over the scheduled plan in 97.7% of cases. Conclusion: This study describes the first workflow to date of a CBCT-based oART combined with a shuttle-based offline approach for MR guidance. The oART duration times reported resemble the range shown by previous publications for first clinical experiences with the Ethos system.

## 1. Introduction

Adaptive radiotherapy (ART) comprises different technologies that aim to reduce the effects of anatomical variation during treatment by periodically adapting the treatment plan to the patient’s current anatomy [1,2]. While systematic errors in anatomic variation, e.g., introduced by the initial planning computer tomography (CT) or progressive changes such as weight loss, can be accounted for by adaptation of the treatment plan with re-imaging between treatment fractions (offline), random variations in anatomy, such as daily variation in organ motion, have to be adapted directly prior to radiation in treatment position (online). By reducing these sources of error, adaptation allows for a reduction in planning target volume (PTV) margins [3,4,5], which potentially decreases toxicity in organs at risk (OAR) in relation to PTV coverage and, in turn, enables isotoxic dose escalation, effectively widening the therapeutic window [6,7,8]. However, practical considerations such as logistics, technical requirements and expertise of personnel limit its widespread adoption [9,10]. In the case of online adaptive radiotherapy (oART), the process needs to be accomplished within a time frame on the order of magnitude of minutes [11,12].

The Ethos radiotherapy system (Varian Medical Systems, Palo Alto, CA, USA) combines a Halcyon-Linac with a streamlined adaptive workflow process designed for rapid treatment adaptation. Daily anatomy changes are detected by kV cone-beam CT imaging. The adaptive workflow utilizes partially artificial intelligence (AI)-based auto contouring of selected OARs as well as Varian’s Intelligent Optimization Engine (IOE) [13] for automated plan generation and adaptation [14]. In practice, previous studies have reported an average duration of plan adaptation sessions on the Ethos system of around 20 min per fraction, yielding on average a total treatment time of 35 min including set-up and radiation delivery [15,16,17,18,19].

In a previous study (MR guidance) in our department, we presented the proof-of-concept workflow and early clinical results of a shuttle-based offline MR-guided radiotherapy system [20,21], which can be a flexible and cost-effective alternative to fully integrated MR–Linac systems. In comparison to the lower-field strength MRI in MR–Linac systems (0.35 T–1.5 T), high-resolution (3T) functional MR imaging in treatment position is integrated for once-weekly offline MR guidance for plan adaptation. Critically, this potentially allows for adaptation to biological response signals, which, especially in the context of positron emission tomography (PET) imaging, has shown better clinical outcomes when integrated into the treatment strategy [22,23]. Growing evidence for the prognostic relevance of quantitative MRI biomarkers implicates their potential utility in determining individual radiation doses [24,25,26].

Our team has developed and deployed a novel approach to online adaptive radiotherapy by utilizing a shuttle-based approach for MR guidance, which we term the modular adaptative radiotherapy system (MARS) (Figure 1). By combining the Ethos system with the shuttle-based approach, we can leverage the benefits of rapid adaptation and advanced treatment planning provided by the Ethos system, while also obtaining the superior soft-tissue contrast and quantitative biological response assessment of MR imaging in treatment position. This approach has the potential to provide more precise and effective treatments for cancer patients, while also minimizing treatment time and reducing the risk of side effects. We here outline the workflow, which, to our knowledge, is the first to explore CBCT-based oART with offline MR guidance, and we report fractional times derived from clinical treatment of 31 patients.

## 2. Materials and Methods

### 2.1. Ethos oART Workflow with Offline MR Guidance

#### 2.1.1. Initial Treatment Planning

The Ethos radiotherapy system contains a 6 MV Flattening Filter Free (6X-FFF) O-Linac and can perform both sliding-window intensity-modulated radiation therapy (IMRT) and volumetric intensity-modulated arc therapy (VMAT). For treatment planning, patients received an initial planning CT and MRI without contrast fluid in treatment position using the shuttle system. If not already available from initial tumor staging, an additional diagnostic contrast-enhanced MRI scan was performed. For contouring and creation of the reference plan, the acquired MR images were rigidly co-registered. Treatment planning was performed using the Ethos treatment planning system (TPS). In the Ethos system, different disease sites are organized into modules for site-specific automatic segmentation of OARs. For semi-automated treatment plan generation, the Intelligent Optimization Engine (IOE) in the Ethos system takes as input an ordered list of dosimetric clinical goals regarding the target volumes and OARs on the baseline CT scan. These clinical goals are then translated into objective functions, which guide the optimization process to generate treatment plans. IMRT plans with 7, 9 and 12 equidistant fields and 2- and 3-arc VMAT plans were generated, among which one was chosen as the reference plan based on pre-set dose constraints. The same beam geometry was used for all subsequent adaptive fractions.

Dose calculations for all plans in the Ethos system were performed using the AcurosXB algorithm (Varian Medical Systems), calculating dose to the median. Plans were normalized to the PTV median dose and were set to achieve at least 95% of the prescribed dose to at least 95% of the PTV (maximum dose 110% of the prescribed dose) and without violation of the OAR goals, as per institutional standard.

#### 2.1.2. Weekly MRI for Offline MR Guidance and Integration into Ethos

Once weekly and directly after the planning CT scan, patients received a short-protocol MRI on a 3T MRI scanner (VIDA, Siemens, Erlangen, Germany) in treatment position, including T2-weighted images and diffusion-weighted images. The mere MRI protocol itself was aimed to undercut 25 min. No contrast fluid was used for these MRI scans.

These MRI scans were integrated into the ongoing radiotherapy by creating a treatment revision. The MRIs were non-deformably co-registered to the CBCT. Direct delineation was performed on the primary CBCT and the MRIs were used to differentiate soft-tissue structures. The image registration itself did not differ from the standard procedure of co-registration of secondary images during creation of the reference plan.

#### 2.1.3. Shuttle-Based Transport System

Transport between the MR scanner and the Ethos as well as between the CT scanner and the MR scanner during treatment planning was conducted in treatment position via a shuttle system (Symphony, CQ Medical, UK). The workflow itself has been described in previous publications [20,21]. Briefly, the shuttle system, which is compatible with both the linear accelerator and the MRI, utilizes hoverboarding technology to transfer the patient on a transport stretcher between the MR or CT scanners and the treatment room, without requiring any movement from the patient (Figure 2). This system uses the same patient platform for all procedures, allowing for the use of treatment supports, immobilization devices and stereotactic tools.

#### 2.1.4. Online ART with the Ethos System

The Ethos oART workflow has been described in detail previously [27]. As outlined in Figure 1, the adaptive workflow is a streamlined process that requires four key decisions to be made by the clinician.

**Figure 2 cancers-16-01210-f002:**
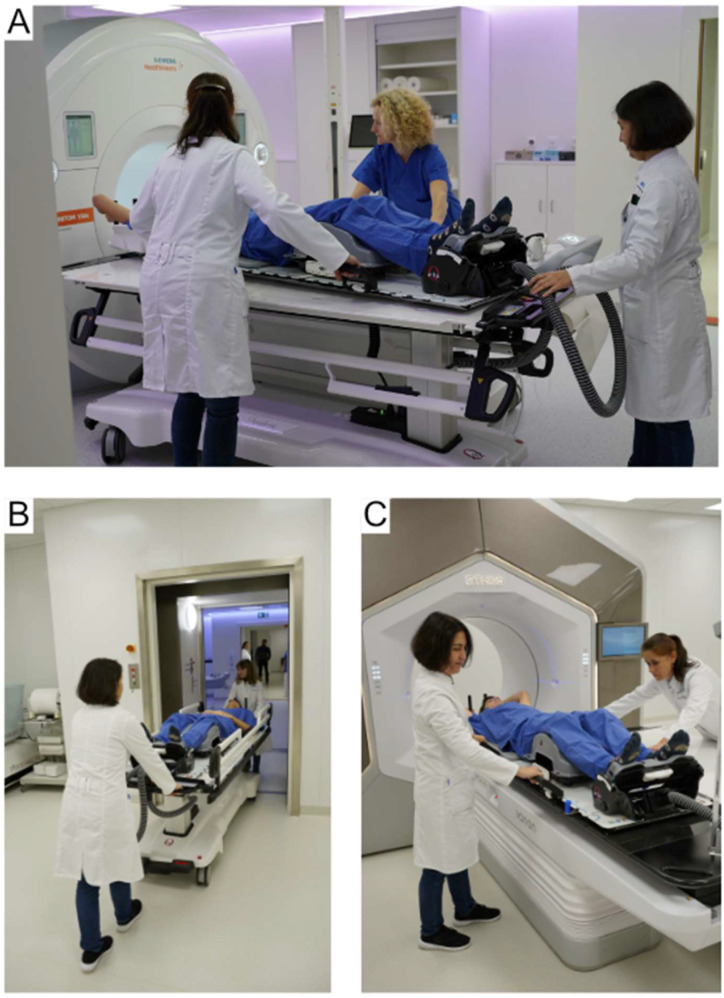
Shuttle-based transport: After acquiring the MRI in treatment position in the 3T MRI scanner (VIDA, Siemens, Erlangen, Germany) (**A**), the patient is transferred to the shuttle in the same position (Symphony, CQ Medical, UK) (**B**) and transported to the Ethos treatment machine (**C**).

Firstly, a kV-CBCT is acquired and, if suitable, accepted for AI-based delineation of influencer structures, which are a group of disease-site-specific OAR. Secondly, influencer structures are reviewed, optionally adjusted and accepted. These structures are used to guide elastic or rigid propagation of target and OAR contours from the reference image to the kV-CBCT. This process creates a synthetic CT that adapts the anatomy of the day while deriving the Hounsfield units from the reference image for later dose calculation. Thirdly, the target and OAR structures are reviewed, adjusted and accepted. In a subsequent step, this session model is utilized by the Ethos treatment planning system (TPS) to generate two plans: a scheduled plan that is re-calculated from the reference plan to the anatomy of the day and an adapted plan that is generated by running a new optimization model by the IOE. Finally, after dosimetric assessment of both plans, the clinician decides which plan should be used for radiation treatment on that day. The choice between the adapted versus the scheduled plan is based on comparisons of dosimetric constraint violations and dose–volume histograms. Further adjustments cannot be made to either plan without manipulation of contours, which would then trigger the calculation of a new adapted plan. After the adaptive procedure but prior to radiation delivery, another verificational CBCT is acquired in cases where it is deemed necessary by physician’s and physicist’s choice.

#### 2.1.5. Physicist QA

Patient-specific quality assurance was performed for both initial and adapted plans using the PTW Octavius phantom and Mobius Dose calculation tools (Varian) for the initial and only Mobius for adapted plans. Evaluation of measured data was performed with Verisoft (PTW) using the 3d gamma method. All generated plans were automatically sent to Mobius for recalculation and evaluation. The QA procedure performed in parallel by the physicist was usually ready when the physician had decided on the use of either the scheduled or adapted treatment plan [28].

### 2.2. Data Acquisition and Analysis

Patient data were derived from the clinical documentation system. Session data including fractional session times were extracted from exported session reports by Ethos using custom scripts. For fractional session time analysis, five different time stamps were used to compute four time phases: Firstly, MRI time was defined from the start of MRI acquisition until the start of kV-CBCT acquisition, therefore including shuttle transport and patient positioning at the Ethos system. Secondly, adaptation time was defined as the time from kV-CBCT acquisition until completion of plan selection. Thirdly, QA time reflected the duration of QA after plan selection and before the start of radiation treatment for adaptive sessions. Finally, RT-to-end time was defined as the beam-on time until the session was manually closed. Total session time was defined as the time from kV-CBCT acquisition until the end of treatment. For scheduled fractions, only MRI time and total session times were available. Notably, the durations computed from the differences in the given time stamps include sources of delay, such as repositioning, bathroom visits and waiting times. These values were not corrected to reflect a realistic clinical setting.

Graphs were computed using the R-package “ggplot2”. Comparative clinical studies were collected from the MEDLINE database via the freely accessible PubMed interface.

### 2.3. Ethics Statement

This retrospective analysis was approved by the ethics committee of the University Hospital Heidelberg (S-511/2023).

## 3. Results

### 3.1. Patients

Patient treatment with this workflow began in February 2023 in our center. Until November 2023, 31 patients completed treatment (Table 1). The most common primary tumor entities were prostate cancer (51.6%, 16 patients), lung non-small-cell lung cancer (22.5%, 7 patients) and cervical cancer (16.1%, 5 patients).

Most patients (87.1%, 27 patients) underwent definitive radiation treatment of their primary tumor without previous radiation treatment in the target volume. Two patients with lung cancer who had undergone previous local chemoradiation with 30 × 2Gy were treated for a local recurrence (Table 2). The other two patients presented with nodal metastasis of prostate cancer after previous radiation treatment and underwent elective nodal irradiation with simultaneous integrated boost of nodal metastases [29]. One patient with prostate cancer received a prolonged treatment with reduced single-fraction doses after sigmorectal extirpation to reduce colon toxicity [30].

Nineteen patients (61.3%) received concurrent or sequential systemic therapy. Among the sixteen patients with prostate cancer, nine received ADT. Patients with definitive treatment of lung cancer, cervical cancer and postoperative endometrial cancer received systemic therapy as per local standard in our institution.

### 3.2. Adaptive Treatment Sessions

In total, 737 fractions were delivered for 31 treated patients (Figure 3, Appendix A). Treatment was completed in all patients.

The median total treatment time without MRI acquisition and transport (kV-CBCT to end of RT) was 30.8 min (IQR 25.0–39.3) for all adaptive sessions (Figure 4, Appendix A). Separately, the median adaptation, plan QA and RT times were 24.4 min (IQR 18.8–32.2), 0.4 min (IQR 0.3–1.0) and 5.3 min (IQR 4.5–6.7), respectively. The number of outliers in adaptation time reflects sessions in which delays or unavoidable interruptions in the workflow were incorporated, e.g., visits to the bathroom during the adaptation process. Furthermore, there were ten fractions in which the RT-to-end time exceeded 10 min (range: 10.0–34.9 min). The beam-on times themselves rarely exceeded three minutes. These outliers reflect manual errors in which the session was closed late and were therefore masked from the figure.

The median total time of those 17 fractions in which the scheduled plan was chosen was 32.2 min (IQR 28.2–37.7) (Appendix A).

The highest median total session time was observed for adaptive sessions in patients undergoing treatment for cervical cancer (43.0 min (IQR 37.3–50.4)), as well as adjuvant radiotherapy for endometrial cancer (40.0 min (IQR 33.6–44.1)) and bladder cancer (38.3 min (IQR 35.3–39.8)), though the latter two treatments involved only one patient each (Figure 5). The lowest median total session time was observed for adaptive sessions of prostate cancer for definitive treatment (27.0 min (IQR 22.1–32.0)).

The adapted plan was chosen over the scheduled plan in 97.7% of cases.

### 3.3. Weekly MR Imaging

In total, 118 MRIs were acquired for offline MR guidance. Weekly MR imaging was omitted in cases where the planning MRI was acquired less than a week before the start of treatment. Due to MRI availability, weekly MRIs were also occasionally omitted in cases where they would be used for fewer than five fractions at the end of treatment. Besides these, there were only two events in which planned MRI sessions could not take place, one of which was due to scheduling conflicts and the other due to refusal by the patient.

The median MRI acquisition time including transport to Ethos was 53.6 min (IQR 46.5–63.4) (Figure 6). Notably, this time included waiting times prior to shuttle transport, such as when the Ethos machine was occupied by a previous patient for treatment or other sources of delay. Furthermore, there were seven sessions in which the computed MRI acquisition time including transport exceeded 90 min (range: 95.5–283.5 min). These represent cases of minor scheduling conflicts and do not reflect the time in which the patients were waiting in treatment position. Therefore, these data points are masked from Figure 6 but were included in the analysis.

## 4. Discussion

We present the first clinical experience of CBCT-based oART with offline MR guidance at the German Cancer Research Center (DKFZ) in Heidelberg, which, to our knowledge, is the first institution to implement, deploy and describe this system.

We compared our fractional session time analysis to other previously published studies in the literature reporting adaptation times (Table 3). The comparison was complicated by differing or missing time measuring points. The two studies by Byrne et al. and Stanley et al. [15,18] report a median total treatment time of around 34 min. Even if we were to account for added time due to patient positioning, which was not included in our metric, we propose that our reported times were in the range of those reported by these studies.

Furthermore, we found three more studies that reported data on only one tumor entity [17,19] or had very low patient and fraction numbers [16] and were therefore not included for comparison.

Regarding MRI times, in comparison to our median MRI + shuttle transport time of 53.6 min, the previous study from our department by Bostel et al. [21] reported median total session times (MRI + shuttle transport + CBCT + RT) of 61 min, in which CBCT with repositioning and RT accounted for 5 and 13 min, respectively, yielding a slightly lower estimated MRI + shuttle transport time of 41 min. There have been other published reports on offline MR guidance. Jaffrey et al. present a mobile MR scanner on a rail system that can be moved between a brachytherapy and a radiotherapy suite [31]. Another report by Nyholm et al. utilized a trolley system that could be attached to the MR and radiotherapy couches and was exclusively used for repositioning [32]. Neither group reported MRI and transport session times.

We can attribute multiple factors as possible sources of delay in our study, most of which, however, are intrinsic and predictable to the concept of our workflow. For instance, the integration of the MRI into the workflow took around 5–10 min. Another contributing factor would have been the trade-off between regular high-resolution information afforded by the acquisition and integration of a weekly MRI and their incentivization for further manual editing. Moreover, having to coordinate patients for the MRI acquisition and their transport to the Ethos machine requires additional resources and is another source of delay, e.g., when the previous patient undergoing radiotherapy had to be finished before the medical technical assistants were ready to transport the next patient to the Ethos machine after the weekly MRI acquisition. Implications to the limitations set to treatable patients, such as the necessity to be able to lie for prolonged periods, have been discussed in previous studies [15].

There are also technical limitations to this workflow. The maintenance of separate imaging and radiotherapy machines in comparison to integrated Linacs may lead to workflow inefficiencies. For example, integrating new MR imaging data into the ongoing Ethos treatment in daily routine is a novel approach and, therefore, not streamlined yet.

Moreover, by nature of its design, real-time imaging during beam-on time cannot be performed in this system, in comparison to the MR–Linac and its cine-MRI feature, which allows for tumor tracking during the radiotherapy session [33,34].

On the other hand, separating each unit has advantages: Firstly, each unit can run independently (e.g., MRI for diagnostics). Secondly, high-quality 3T MR images can be obtained, enabling thorough morphological and biological soft-tissue evaluation. There are emerging data on functional MRI biomarkers prior to and during radiotherapy with potential prognostic relevance [35,36,37]. We believe these to be crucial advantages in favor of modular systems in order to heighten affordability and utilization.

While the notion of oART is clearly intuitive, there has so far been limited prospective clinical evidence in support of oART regarding efficacy in terms of local control or superior toxicity profiles, as reviewed elsewhere [10]. Moreover, safety concerns remain about certain aspects of ART, such as adapting to tumor shrinkage leading to possible underdosage of micrometastases in the adapted tissue volume [38], with corresponding differences in ART implementation between studies. Further studies are required to prove whether ART is clinically effective. Crucially, for ART to be considered cost-effective, feasible ART solutions must be developed that demonstrate substantial and noticeable improvements in clinical studies.

Further dosimetric and long-term studies will follow to investigate the clinical benefits of our workflow. Within the scope of this study, we managed to show that the workflow is feasible without major disruptions in the workflow. In a next step, relevant questions addressed by these studies would be to identify patient groups that benefit most from the adaptive process, as well as to identify clinical or morphological biomarkers that predict the benefit of oART and/or offline MR guidance.

Currently, clinical trials are underway to further validate the feasibility and efficacy of the presented workflow and to explore its potential benefits for a range of cancer types, such as cervical cancer within the AIM-C1 trial (daily AI-based treatment adaptation under weekly offline MR guidance in chemoradiotherapy for cervical cancer) [39].

## 5. Conclusions

CT-guided daily oART with weekly, offline MR guidance using a shuttle-based system to transport the patient between a 3T-MRI and the Ethos radiotherapy machine in treatment position is feasible and combines the benefits of both daily oART and superior soft-tissue resolution from the MRI.

## Figures and Tables

**Figure 1 cancers-16-01210-f001:**
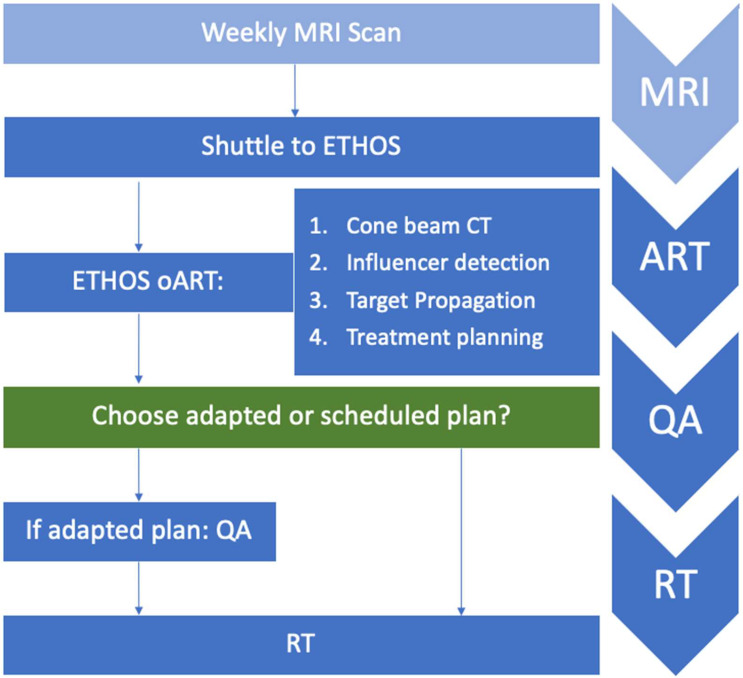
CBCT-guided online adaptive radiotherapy (oART) with magnetic resonance (MR) guidance workflow: During treatment, patients underwent MR imaging in treatment position for offline MR guidance once a week (light blue). Subsequently, they were shuttled in the same (treatment) position to the Ethos radiotherapy device. Four decisions are made in the oART process, which are outlined in Section 2.1.4. Briefly, a kV-CBCT was acquired to capture the anatomy of the day by artificial intelligence-supported segmentation of organs at risk and target structures. These are used to guide deformation of the planning CT, which in turn is used to generate a scheduled and an adapted plan. If the adapted plan is chosen for the treatment of the day, plan-specific quality assurance (QA) is performed prior to initialization of radiation treatment. The arrows on the right refer to the computed fractional session times of the treatment sessions, as defined and outlined in Section 2.3.

**Figure 3 cancers-16-01210-f003:**
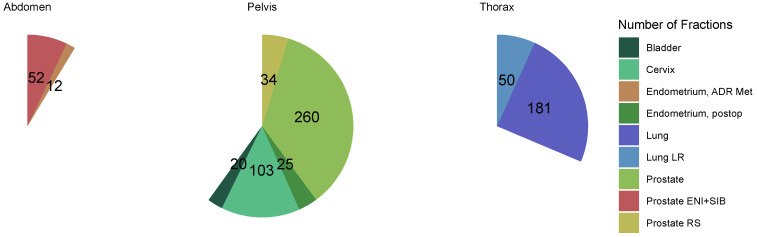
Number of fractions delivered by entity and treatment region (*n* = 737). Treatment sites to which fractions were applied were divided into three regions: abdomen (*n* = 64), pelvis (*n* = 442), thorax (*n* = 231). Entities refer to the different treatment concepts listed in Table 2. Abbreviations: ADR Met: adrenal gland metastasis, ENI + SIB: elective nodal irradiation + simultaneous integrated boost, LR: local recurrence, RS: rectal surgery.

**Figure 4 cancers-16-01210-f004:**
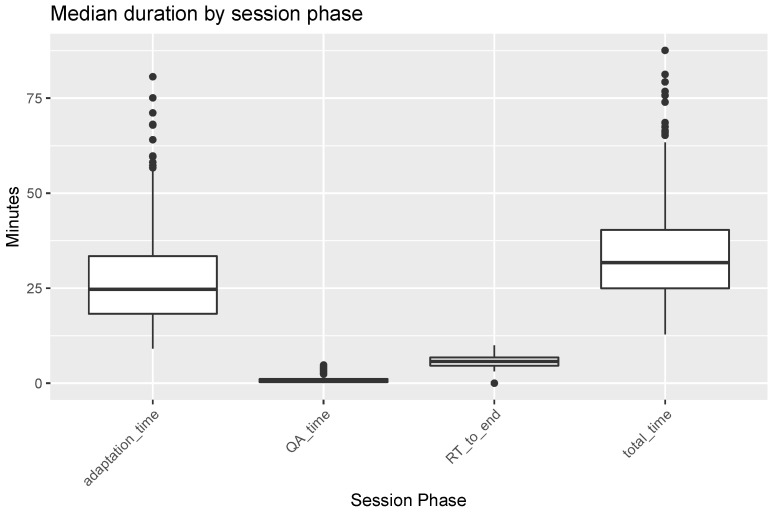
Median duration of fractional session times in all adaptive sessions (*n* = 720) on the Ethos radiotherapy machine. Time definitions are defined in Section 2.3. Briefly, adaptation time refers to the time from CBCT to plan selection, including adaptation. This is followed by QA if an adapted plan is chosen. RT-to-end time refers to the time directly following QA until the session is closed, including beam-on time. Outliers > 10 min (*n* = 10) in the RT-to-end column are not shown but were included in the analysis. Total time refers to the sum of these three phases for each adaptive session.

**Figure 5 cancers-16-01210-f005:**
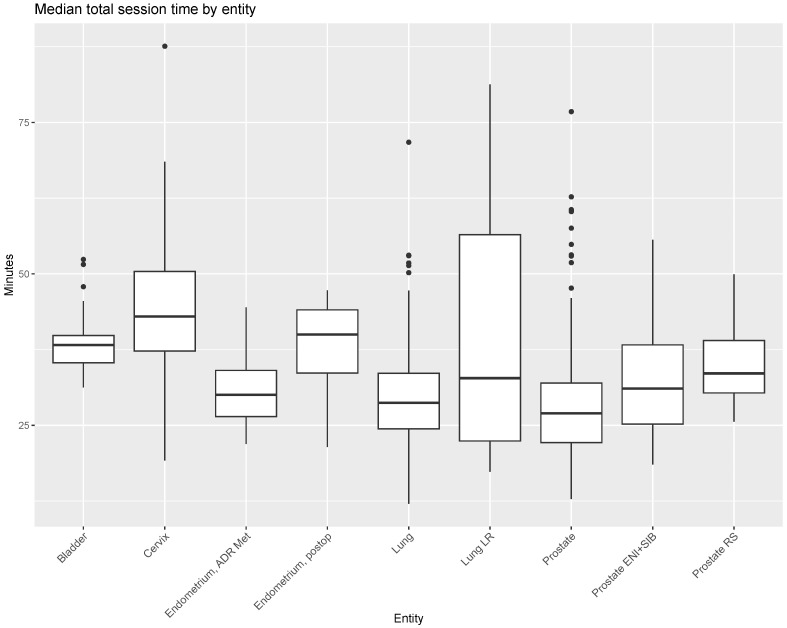
Median total session time of Ethos oART adaptive sessions by treatment intent. Entities refer to the different treatment concepts listed in Table 2. The number of fractions applied were bladder (*n* = 19), cervix (*n* = 103), endometrium postop (*n* = 25), endometrium ADR Met (*n* = 11), lung (*n* = 170), lung LR (*n* = 50), prostate (*n* = 257), prostate ENI+SIB (*n* = 52) and prostate RS (*n* = 33). Abbreviations: ADR Met: adrenal gland metastasis, ENI + SIB: elective nodal irradiation + simultaneous integrated boost, LR: local recurrence, RS: rectal surgery.

**Figure 6 cancers-16-01210-f006:**
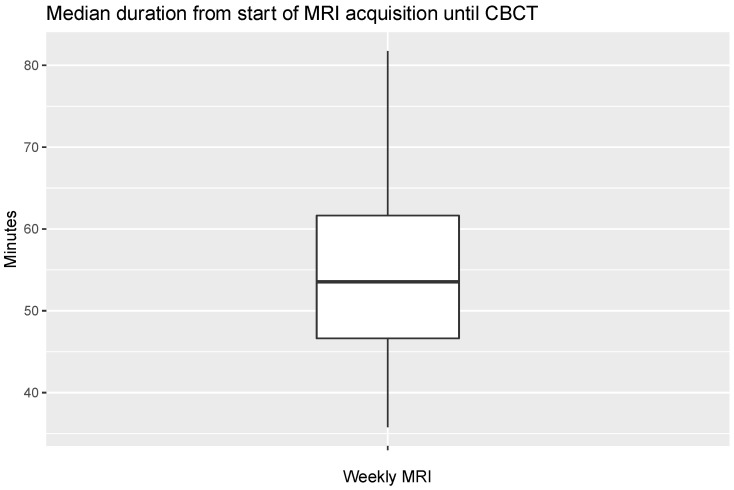
Median duration of weekly acquired MRIs for offline MR guidance (*n* = 118). Outliers > 90 min (*n* = 7) are not shown but were included in the analysis. Time measured from start of MR acquisition includes shuttle transport and positioning until kV-CBCT at the Ethos machine.

**Table 1 cancers-16-01210-t001:** Patient characteristics.

	Median (IQR)
*Age* (*years*)	70 (65–75)
*Karnofsky performance score* (*%*)	90 (80–90)
	**Number of patients (*n* = 31)**
*Sex*	
Male	18
Female	13
*Primary Tumor Site*:	
Prostate	16
Lung	7
Cervix	5
Endometrium	2
Bladder	1
*Treatment Site*:	
Thorax	7
Abdomen	3
Pelvis	21
*Previous RT* (*80% isodose*) *overlapping with target volume*:	
No	27
Yes	4
*Systemic Therapy*:	
None	12
ADT	9
Preceding, sequential chemotherapy	6
Concurrent chemotherapy	5

Abbreviations: ADT = androgen deprivation therapy.

**Table 2 cancers-16-01210-t002:** Treatment concepts by primary tumor.

Primary Tumor	Dose Concept	*n* = 31	Total Fx	Target
Prostate	20 × 3.0 Gy	13	260	Prostate, definitive
	34 × 2.25 Gy	1	34	Prostate, after rectal surgery
	26 × 1.8 Gy ENI; 26 × 2.2Gy SIB	2	52	ENI + SIB
Lung	30–33 × 2.0 Gy	5	181	Lung, definitive
	25 × 1.8 Gy	2	50	Lung, local recurrence
Cervix	25–28 × 1.8 Gy + BT	5	103	Cervix, definitive
Endometrium	25 × 1.8 Gy + BT	1	25	Pelvis, postoperative
	12 × 4 Gy	1	12	Adrenal metastasis
Bladder	20 × 2.75 Gy	1	20	Bladder

Abbreviations: BT: brachytherapy, ENI: elective nodal irradiation, Fx: fractions, SIB: simultaneous integrated boost.

**Table 3 cancers-16-01210-t003:** Comparative studies reporting average durations of Ethos adaptive sessions and MRIs for offline MR guidance.

Study	Patients	Fractions	Entities	Ethos Total Treatment Time
Current study	31	720	Mult.	Median 30.7 min (IQR 24.7–39.2) ^†^
Byrne et al. [18]	6	184	PC	Avg. 34.2 min (±SD 6.6) ^‡^
Stanley et al. [15]	97	1667	Mult.	Avg. 34.5 min (±SD 11.4) ^‡^

^†^ Time from CBCT to end of RT. ^‡^ Time from patient set-up to end of RT. Abbreviations: PC: prostate cancer.

## Data Availability

The data presented in this study will be available on reasonable request from the corresponding author.

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
