# Peer review of "Clinical Workflow of Cone Beam Computer Tomography-Based Daily Online Adaptive Radiotherapy with Offline Magnetic Resonance Guidance: The Modular Adaptive Radiotherapy System (MARS)"

_cancers, 2024, doi:10.3390/cancers16061210_

Round 1
Reviewer 1 Report
Comments and Suggestions for Authors
The authors present in this work the first workflow experience on a combination of CBCT-based oART and diagnostic MRI with an on rail system. Whereas the clinical workflow and the corresponding experience is definitively of interest for the community, key elements of the method are missing and it mainly focuses only on the time needed for oART adaptation. Therefore, an more detailed inclusion of main workflow elements within the methods would improve the manuscript.
1. Were the MRIs used only on the day they were acquired within the online workflow, or were the weekly MRIs as well used on the following fractions?
2. How and how long was the workflow to integrate the MRI images into the TPS after acquiring?
3. Was the registration of the MRI to the CBCT evaluated within the online workflow? If the developed method sufficiently minimizes motion.
4. Within the online workflow , especially i.e. for prostate, the understanding of the current system is that the set automatic registration for oART can not be changed during adaptation. Is there an additional change of registration possible within the oART workflow or must the automatic registration be used as is? How was your process on online adaptation? Match CBCT- MRI tumor and delineate then on the MRI until the next MRI?
5. You note, that you have treated quite some lung patients adaptively on the Ethos system. However there seem to be some concerns on this behalf. (i.e. https://pubmed.ncbi.nlm.nih.gov/38386919/) Did you take additional measures for this entities?
Figure 4: There is a RT time of ~35 minutes noted. Is this a correct irradiation time?
Line 55: “Offline MR.guidance was well tolerated” did you have a questionnaire for this?
Line 77/78 what is meant with adjacent to the tumor?
Methods: Which CBCT/Ethos TPS system is used ? Hyperisght?
Line 140: Plans were generally prescriped to the median dose ?
Line 185: Were there defined criteria for re-CBCT or was it a physicians choice?
Line 188ff: If a revision is done for the inclusion of the MRI, a new “reference” plan must be generated each time? Was this reference plan each time newly measured/QA acquired after the MRI inclusion?
Figure 3: This figure seems quite confusing. Maybe the number of Fx could be included in table 2.
Figure 6: Is the information on the outliers given somewhere? If not they should be shown in figure as well.
Table 3: What does the * and ** reference to?
Comments on the Quality of English Language
Quality is sufficiant.
Reviewer 2 Report
Comments and Suggestions for Authors
The manuscript by Kim et al describes a retrospective experience in treatment of patients on a combined Ethos adaptive radiotherapy linac with an shuttle-linked diagnostic 3T MRI machine. The manuscript is well written and citations are appropriate in this emerging field. There are several additions to the manuscript which are necessary prior to publication.
1. The authors describe a potential avenue in improvements in clinical outcomes by reducing PTV margins - was this done routinely and if so, can this be referenced in the most commonly treated disease sites (prostate/lung/cervical)?
2. On line 261 the authors reference that the adaptive plan was chosen in 98% of the treatments. The criteria used to choose an adaptive plan needs to be described. Was there coverage thresholds for the targets or dose violation for organs at risk? In the discussion there should be mention of the specific future directions the authors like to explore in assessing the dosimetry benefits in these adaptive plans.
3. It is curious that the weekly diagnostic MRI imaging is described but there is no reference to the the clinical implications that this additional imaging allows for. This is especially important due to the patient time and resources used to obtain these data. There is emerging data in weekly MRI assessment of disease response to radiotherapy in many disease sites - can the authors reference these data? Furthermore in context of this manuscript, the authors should discuss the clinical benefit of weekly MRI in the adaptive portion of the workflow. How many cases were re-planned due to weekly MRI changes? Were these variations also evidence on kV-CBCT and would this mitigate some of the benefit of the MRI?
Minor Comments:
1. Line 27: "tumor tissue" - "cancerous tissue." unusual wording.
2. Would recommend only variations of dark colors be used in the pie chart as well as verify formatting on final manuscript.
3. Figure 1 - The figure may be improved by inclusion of the median time and range for each step in the workflow.
4. Line 334: "direct monitoring" - "tumor tracking"
Reviewer 3 Report
Comments and Suggestions for Authors
Dear authors,
This works represents a great research about the development of new approaches using MRI for understand a serious and relevant disease, as cancer. I really enjoyed the manuscript, which is well written and designed. The figure are in great resolution. The results are well present, and mainly in the discussion section, it is provided a great explanation about the results and some challenges faced by the authors. Finally, the conclusion section is very objective and summarize the main finding of the paper.
I would like to do some minor considerations, mainly about personal curiosities?
1-) Is there any plan to application on studies regarding others relevant diseases, as Alzheimer or other study patients that have other kind of cancer, as pancreas?
2-) What is the perspective of these studies, regarding the monitoring of tumoral progress activities? Do you believe that these methods can be used to understand and propose early diagnosis?
Thanks loads,
Round 2
Reviewer 1 Report
Comments and Suggestions for Authors
The authors have sufficiantly answered my question and increased the manuscript quality.
Reviewer 2 Report
Comments and Suggestions for Authors
My questions were adequately addressed.